# Circadian Clock Desynchronization and Insulin Resistance

**DOI:** 10.3390/ijerph20010029

**Published:** 2022-12-20

**Authors:** Federica Catalano, Francesca De Vito, Velia Cassano, Teresa Vanessa Fiorentino, Angela Sciacqua, Marta Letizia Hribal

**Affiliations:** Department of Medical and Surgical Sciences, University of Catanzaro “Magna Graecia”, 88100 Catanzaro, Italy

**Keywords:** circadian clock, insulin sensitivity, insulin signaling, peripheral tissues

## Abstract

The circadian rhythm regulates biological processes that occur within 24 h in living organisms. It plays a fundamental role in maintaining biological functions and responds to several inputs, including food intake, light/dark cycle, sleep/wake cycle, and physical activity. The circadian timing system comprises a central clock located in the suprachiasmatic nucleus (SCN) and tissue-specific clocks in peripheral tissues. Several studies show that the desynchronization of central and peripheral clocks is associated with an increased incidence of insulin resistance (IR) and related diseases. In this review, we discuss the current knowledge of molecular and cellular mechanisms underlying the impact of circadian clock dysregulation on insulin action. We focus our attention on two possible mediators of this interaction: the phosphatases belonging to the pleckstrin homology leucine-rich repeat protein phosphatase family (PHLPP) family and the deacetylase Sirtuin1. We believe that literature data, herein summarized, suggest that a thorough change of life habits, with the return to synchronized food intake, physical activity, and rest, would doubtless halt the vicious cycle linking IR to dysregulated circadian rhythms. However, since such a comprehensive change may be incompatible with the demand of modern society, clarifying the pathways involved may, nonetheless, contribute to the identification of therapeutic targets that may be exploited to cure or prevent IR-related diseases.

## 1. Introduction

Insulin resistance (IR) is a broad term encompassing any condition showing an impaired response to the hormone insulin in its target tissues [1]. According to the so-called common soil hypothesis, IR represents the nurturing ground for several diseases, including type 2 diabetes, obesity, hyperlipidemia, fatty liver disease hypertension, and cardiovascular complications [2]. The prevalence of these diseases has been constantly increasing, with ~500 million adults affected by type 2 diabetes worldwide in 2021 [3].

Insulin, an endocrine peptide hormone secreted by pancreatic β-cells, is the master controller of carbohydrate, lipid, and protein metabolism. Upon food ingestion, glycemia rises, insulin release is stimulated, and the secreted hormone promotes several tissue-specific responses. These include the suppression of endogenous hepatic glucose production (HGP); the stimulation of glucose uptake by muscle, liver, and adipocytes; the inhibition of lipolysis; and vasodilation in skeletal muscle, which further favors glucose disposal [4].

Insulin signaling is triggered by its binding to the transmembrane insulin receptor (InsR). This activates the InsR intrinsic tyrosine kinase activity and promotes tyrosine phosphorylation of InsR substrate (IRS) proteins, thus providing docking platforms for proteins that contain *src* homology region 2 (SH2) domains. Two fundamental downstream signaling pathways are activated upon IRS proteins binding to the InsR: the phosphatidylinositol-3-kinase (PI3K) pathway and the mitogen-activated protein kinase (MAPK) pathway. PI3K signaling promotes the phosphorylation of key downstream effectors, including the serine/threonine kinase Akt, also known as protein kinase B (PKB), and its substrates, involved in both metabolic and pro-survival insulin actions. The activation of the MAPK cascade plays a critical role in the insulin-mediated regulation of mitogenic events such as cell proliferation and differentiation, as well as survival [1]. Insulin signaling is negatively governed by numerous mechanisms such as the binding of the adaptor protein Grb10 to the kinase domain of InsR; serine phosphorylation of IRS proteins; dephosphorylation of InsR and IRSs by protein tyrosine phosphatase-1B (PTP-1B); or Akt inhibition mediated by the pseudo-kinase tribbles homolog (TRIB3) or by the pleckstrin homology leucine-rich repeat protein phosphatase family (PHLPP) [5,6].

The circadian rhythm consists in an endogenous variation of biological processes that occur within 24 h in living organisms such as animals, plants, fungi, and bacteria [6,7,8]. These endogenous rhythms reflect the functions of an intrinsic circadian clock which mediates physiological and behavioral processes with the aim of coordinating the internal environment with the external one. The term biological clock was coined to identify the daily changes caused by continuous environmental variations to improve the survival of all existing organisms [7,8,9]. The biological clock controls the activation of intracellular signaling pathways; cell proliferation; DNA damage repair and response; angiogenesis; metabolic and redox homeostasis; and inflammatory and immune response of several tissues and organs [7,8,9]. The circadian timing system is characterized by a central clock in the suprachiasmatic nucleus (SCN) and peripheral clocks in peripheral tissues, including muscle, fat, and liver [7,8,9]. The SCN receives the light signal through the retina and the retinohypothalamic tract, and this is the most important activating signal or “Zeitgeber” (ZT) for the SCN. The molecular regulation of the central and peripheral clocks is based on transcriptional-translational feedback circuits. The clock genes involved in these circuits are neuronal PAS domain-containing protein 2 (NPAS2); brain and muscle ARNT-like protein 1 (BMAL1); the period circadian regulator genes (PER1, PER2, and PER3); Cryptochrome Circadian Regulator (CRY1 and CRY2) genes; the genes encoding nuclear receptor subfamily 1 group D (NR1D1 and NR1D2); and retinoid-related orphan receptor (ROR A, B, and C). The central clock communicates with peripheral tissues through neural, endocrine, temperature, and behavioral signals and consequentially each one responds autonomously to the central clock [7,8,9]. Circadian rhythm alterations occur with a high frequency, with approximately 20% of individuals involved in shift work, 33% of the worldwide population sleeping no more than 6 h per night, and 69% experiencing social jet lag [10].

Glucose homeostasis in healthy subjects shows circadian rhythmicity. In the morning before the feeding/active period, insulin sensitivity is greater, fasting insulin levels are higher, and the initial phase of insulin secretion is enhanced as compared to the evening, before the fast/inactive period [11,12].

Consequently, as confirmed by several epidemiological studies [6,7,8], a desynchronization of central and/or peripheral clocks is associated with an increased incidence of IR and related diseases. Here we will focus on the possible mechanisms underlying the impact of circadian clock dysregulation on insulin action.

## 2. Impact of Central and Peripheral Clocks Misalignment

The first, and more straightforward, explanation for the association between circadian rhythm dysregulation and impaired insulin action may lie in the impact of central and peripheral clocks misalignment. A preserved insulin action in the brain is needed to maintain whole-body glucose homeostasis. Skinner et al. demonstrated that in mice subjected to twelve 6 h phase shifts in a protocol efficaciously mimicking shiftwork schedules, insulin was unable to reduce the number of cells carrying an IRS-1 form phosphorylated at the inhibitory serine 612 residue. In the same animals, a compensatory enhanced activation of Akt was observed as compared to control mice not exposed to light cycle disruption [13]. In addition, peripheral clock desynchronization may also result in reduced tissue metabolic flexibility, i.e., the capacity to adapt to changes in fuel availability [14], independently from central effects. Importantly, it was reported that gastrointestinal (GI) functionality is clock controlled to guarantee synchrony with food availability [14]. It is not only gut motility, which defines the transit time of nutrients throughout the GI tract, that possesses circadian rhythmicity, but the ability of GI organs to digest and absorb nutrients also appears to be modulated according to intrinsic and extrinsic circadian rhythms. Thus, the expression of specific enzymes involved in the digestion of the three macronutrient categories peaks during the feeding period in anticipation of food intake. Subsequent nutrient absorption is also under circadian regulation, with increased expression of hexose, peptide, and lipid transporters during the feeding period (Table 1) [14,15,16,17]. Notably, the rhythmic oscillations in the expression levels of some of these transporters, such as the sodium-dependent glucose transporter 1 [18], may have a role in determining the rhythmicity of gut hormone secretion, which will be discussed below.

Nutrient utilization at the level of peripheral tissues also follows a circadian pattern. Importantly, studies in animal models suggest that dysregulated feeding cycles interfere with hepatic metabolic genes oscillations, resulting in a desynchronized release of metabolites in the circulatory system [19]. Liver-specific BMAL1 knockout mice have attenuated insulin sensitivity during the fasting phase, while hepatic overexpression of E4bp4, the transcription factor driving PER2 circadian profile, induces marked IR in the liver and in skeletal muscle, associated with reduced fatty acid oxidation during the inactive phases [20]. The unbalanced flux of hepatic metabolites, including free fatty acids and glucose itself, toward the adipose tissue (AT) or the skeletal muscle may activate noxious intracellular pathways that inhibit insulin signaling activation (Table 1) [21,22]. In addition, dysregulated circadian homeostasis may dampen the insulin sensitizing potential of physical activity. It has, in fact, been observed that muscle clock players trigger different metabolic responses to exercise according to the time of the day [23]. Thus, desynchronized meal and exercise timing would have an additive effect resulting in worsened insulin sensitivity (Table 1).

An additional indirect mechanism by which circadian rhythm misalignment may alter insulin sensitivity is through its impact on adipose tissue biology. Circadian clocks not only regulate adipose tissue metabolism in keeping with the effects on other metabolic tissues, as described above, but they also gate the multiday process of adipocyte differentiation. Particularly, in vitro studies showed that preadipocytes irreversibly differentiate in repeated daily bursts, occurring during the resting phase in humans, and are restricted to a short time-window defined by a switch from low-to-permanently high levels of the transcription factor peroxisome proliferator activated receptor gamma (PPARγ), the master regulator of fat cells differentiation [24]. Environmental clock disruption, induced by nightshift work or by the need to adapt to different time zones, may confine the cellular differentiation process to a wrong phase, thus impairing the coordination with other circadian-regulated systems, such as metabolism, DNA repair, and cell proliferation. Moreover, massive cell differentiation commitment to the wrong phase could make adipogenesis uncontrollable [24]. In mice, environmental light-induced clock dysfunction, mimicking rotating shiftwork schedule, results in both visceral and subcutaneous accumulation of fat depots along with alterations in the adipocytes transcriptome with an upregulation of adipogenic, pro-inflammatory, and angiogenic-related genes. These changes are associated with impaired insulin signaling coupled with suppression of the mTOR signaling pathway [25].

The inner clock of differentiated adipocytes also regulates their secretory pattern-discussed in more detail below as well as their whole inflammatory profile. In omental fat cells of obese subjects, it has been observed that the nuclear factor kappa-light-chain-enhancer of activated B cells (NF-kB) induces inner clock system disruption, competing with BMAL1 for the binding of PER2 and chemokine CCL2 promoter regions. This results in decreased PER2 expression, paralleled by a lengthening of the circadian period, and also in increased CCL2 chemokine levels, which exert a critical role in adipose inflammation and IR. Overall, BMAL1 chromatin binding usually occurs in close proximity to NF-κB consensus motifs. Thus, the latter increased overexpression or activity, causes a reprogramming of the trascriptomic profile of omental fat cells with enrichment for factors involved in endocytosis, proteoglicans, and Forkhead Box protein O (FoxO) signaling. These data were confirmed ex-vivo in adipocytes collected from high fat fed obese mice. Interestingly, circadian dysfunction precedes the upregulation of pro-inflammatory factors, suggesting that inner adipocyte clock disruption occurs very early in the obesity development process and may have an important role in the pathogenesis of IR (Table 1) [26].

## 3. Impact of Circadian Secretory Profiles

As already reported in the previous paragraph, the circulatory levels of many hormones and secretory products follow circadian rhythms. Several of these molecules affect insulin action directly or indirectly. Notably, the cell-autonomous clock within pancreatic beta cells anticipates the start of the active feeding period with a peak in insulin secretion [27]. However, if the expected nutrient flux does not arrive, high circulatory insulin levels acutely cause hypoglycemia, but may also in the longer term result in a downregulation of insulin receptor levels [28], thus contributing to IR. Specularly, both insulin secretion and peripheral insulin sensitivity decline at the end of day in anticipation of sleep. If food is consumed at this time, it cannot be adequately metabolized and this would result in nutrient overload as exemplified above. The release of the counterregulatory hormone glucagon [29] as well as of the gastrointestinal glucagon-like hormones should also be carefully timed. Released into the circulation upon food intake, glucose-dependent peptide (GIP) and glucagon-like peptides (GLP-1 and GLP-2) account for approximately 50–70% of the insulin response to nutrient ingestion. Numerous studies have shown that GLP-1 secretion peaks at the beginning of the feeding period while being reduced in conjunction with fasting, suggesting that food intake may be the primary ZT for intestinal L-cells to produce GLP 1 (Table 1) [15]. In addition to those with a direct impact on insulin release, the clock-controlled secretion of other hormones may also contribute to the modulation of insulin sensitivity. In fact, accruing evidence supports the hypothesis that cortisol, the main endogenous glucocorticoid (GC), may be involved in the circadian rhythm disruption-mediated insulin action impairment. The circadian clock system regulates the secretion and the response to endogenous GCs by inhibiting GCs receptor function [30]. Simultaneously, GCs have been found to be implicated in the synchronization of clock gene expression [31]. GCs directly regulate glucose metabolism in different ways, including the well-known interference with insulin receptor signaling in the skeletal muscle [32]. Moreover, in C_2_C_12_ myotubes treatment with non-physiological hydrocortisone (HC) concentration during bathyphase, the time-point of circadian rhythm corresponding to the early evening, is associated with an impairment of insulin sensitivity coupled with the downregulation of genes involved in the intracellular insulin receptor signaling and fatty acid metabolism, including *Insr, Irs1, Irs2, Pi3kca,* and *Adipor2.* Conversely, non-physiological or physiological HC concentrations during acrophase and midphase, corresponding to the early morning and the early afternoon, respectively, do not affect insulin sensitivity [33]. This observation hints to a putative mechanism by which dysregulated circadian rhythm may negatively affect insulin action.

Another important hormone involved in the regulation of insulin sensitivity is ghrelin, a stomach-derived hormone released in a diurnal oscillation with increased levels during fasting and low levels during feeding phases [34]. It promotes food intake and fat accumulation resulting in weight gain and adiposity, both in humans and in animal models, thus playing a crucial role in the development of IR-related disorders [35]. Interestingly, more recent in vivo studies have demonstrated that intestinal ghrelin increases HPG by activating a gut-brain-liver neurocircuitry and is also able to downregulate insulin secretion ex-vivo in human islets [36]. Moreover, it inhibits insulin signaling by reducing InsR and Akt phosphorylation levels [37].

As detailed above, the circadian clock is important for adipocyte differentiation; however, it also regulates rhythmic synthesis and the release of adipose tissue-specific small polypeptides, collectively known as adipokines [38]. Among these molecules, leptin acts as a satiety hormone counteracting food intake by downregulating orexigenic neuropeptides *Agrp* and *Npy* expression and promoting the expression of anorexigenic *Pomc*. This adipokine also regulates energy expenditure increasing the expression of the uncoupling protein 1 gene (*Ucp1)* in brown adipose tissue (BAT) and activating β-oxidation in peripheral tissue via the AMPK pathway, thus preventing lipid accumulation in these tissues. Finally, it directly inhibits GCs release which, as outlined above, are involved in glucose and lipid homeostasis [39]. Visfatin is another important metabolic regulator encoded by the *Nampt* gene with a diurnal rhythm that is inversely related to that of leptin. Visfatin exerts pro-inflammatory properties; in fact, it has been suggested that it induces IR in the liver in part by promoting an inflammatory state [39,40].

Furthermore, alike adiponectin, the most clinically relevant cytokine associated with obesity exhibits a rhythmic expression in AT. As a downstream effector of fibroblast growth factor 21 (FGF21), adiponectin plays a critical role in the regulation of energy metabolism and insulin sensitivity [41]. Adiponectin receptors are expressed in a circadian manner not only in AT, but also in the mediobasal hypothalamus where adiponectin conveys the peripheral metabolic state to the brain, thus regulating food intake [42]. Adiponectin action is tightly balanced by resistin, another adipokine released by AT, as well as by resident macrophages, whose expression levels negatively correlate with gastric content and serum insulin concentration. Once released, resistin promotes pro-inflammatory cytokines expression thus contributing to IR and inflammation [38]. An additional circadian-controlled adipokine is retinol-binding protein 4 (RBP4). A mouse study has demonstrated that plasma, as well as tissue, RBP4 levels oscillate across the circadian light/dark cycle, with the liver in fact being the preeminent production site, followed by adipocytes. Interestingly, liver specific RBP4 ablation significantly improves insulin sensitivity in mice at ZT4, but has no effect at ZT16, suggesting that RBP4 has a role in modulating glucose metabolism mainly at fasting time (daytime in mouse models) [9]. In a subsequent study, adipocyte-derived RBP4 was also demonstrated to modulate insulin sensitivity at specific time points by interacting with STRA6, a receptor expressed in white adipose tissue (WAT) and skeletal muscle. It then activates the JAK/STAT pathway, which in turn is responsible for the transcription of genes such as Socs3, a potent inhibitor of insulin receptor responses [43]. STRA6 mRNA expression also shows circadian oscillations with a significant peak observed at ZT22-2 when mice are known to be relatively insulin-resistant, and a nadir is observed at ZT14 (Table 1) [44].

Finally, the quintessential circadian rhythm regulator hormone, melatonin, may also be implicated in mediating the impact of disrupted circadian regulation on peripheral insulin sensitivity. Melatonin promotes energy consumption, limiting WAT accumulation and thus indirectly enhancing insulin sensitivity [45]. A recent study in a mouse model of circadian disruption induced by 24 h light exposure, showed that melatonin supplementation ameliorates IR, improving intestinal lipid absorption and digestion. Interestingly, melatonin also attenuated gut dysbiosis. Since microbiota alterations, as detailed below, may have a role in IR, it is possible to hypothesize an additional interconnection between these regulatory mechanisms [46]. A more direct effect on intracellular insulin signaling may also be hypothesized since it was reported that melatonin activates the PI3K/Akt pathway in mice lacking melatonin receptor 1 (MT1). Furthermore, they show desynchronized expression of all PI3K subunits (p50α, p55α, p85α, and p110α) and consequently, an impaired assembly of the active enzyme that results in reduced activation of the downstream signaling cascade [47]. 

## 4. Role of the Microbiome

The intestinal microbiome and the various forms of microorganisms that populate the gastrointestinal tract have recently been recognized as possible regulators of the circadian rhythm [48]. Several studies conducted on BMAL1 and PER1/2-deficient mice have shown a close correlation between the host circadian clock and the microbiome. Sequential sampling of the microbiome, performed on these animal models, demonstrated a significant loss of rhythmicity and reduced number of cyclic microbial species [49]. The microbial ecosystem can undergo different pressures derived from the rhythmicity of the immune system, intestinal epithelial rhythms, and rhythmic metabolic activity. The microbiome regulates rhythmic gene expression in different organs both through direct recognition of microbial molecules and through the release of metabolites [50]. The microbiome also influences diurnal fluctuations of certain classes of hormones and their receptors [48], thus indirectly impacting insulin sensitivity as described above. The circadian rhythm and the microbiome are, in fact, capable of positively influencing but also disrupting each other. Short-chain fatty acids (SCFA) produced by resident microbial species, including butyrate, propionate, and acetate, play an important role in diet-induced obesity and IR. SCFAs regulate the activation of various pathways involved in cholesterol, lipid, and glucose metabolism. Furthermore, several studies on animal models have shown how alterations in the circadian rhythm led to reversible changes in the microbial community and consequently, through the interruption of the colon epithelium barrier, to the activation of a systemic inflammatory state and consequently to a reduction of insulin sensitivity [51].

## 5. Candidate Mediator Molecules

An attractive putative mediator for the impact of circadian rhythm dysregulation on insulin sensitivity may be the NAD+-dependent deacetylase Sirtuin 1 (SIRT1). SIRT1 is a class III histone deacetylase protein and belongs to a family comprising seven members (SIRT1-SIRT7) with different functions and intracellular localization. Sirtuins are key players in several age-related disorders, including cancer, neurodegeneration, and cardiovascular diseases [52]. Specifically, SIRT1, which shows a nuclear localization, was implicated in the modulation of genes responsible for the control of metabolism and was demonstrated to be able to promote insulin sensitivity, counteracting the detrimental impact of chronic inflammation [53]. Interestingly, SIRT1 expression, as well as its activity, shows a circadian pattern, with maximal and minimal levels reached at around ZT16 and ZT4, respectively, in mouse liver [54]. Specularly, SIRT1 modulates circadian rhythm as it can deacetylate BMAL1, thus disrupting *NPAS2*/BMAL1 complex [55] and PER1 accelerating its degradation in the proteosome [55]. The hypothesis that SIRT1 may have a key role in both insulin sensitivity and circadian rhythm pathways found experimental support in the observation that in cultured C_2_C_12_ myotubes with SIRT1 ablation, *NPAS2* and BMAL1 overexpression failed to attenuate palmitate-induced IR. By contrast, ectopic expression of SIRT1 improved IR, induced by the knock-down of BMAL1 or *NPAS2* in the same cellular model [56]. Similar results were obtained in a previous study, investigating insulin sensitivity in liver specific *Bmal1* knock-down mice [57]. 

Mechanistically, an explanation for SIRT1 ability to positively modulate both insulin sensitivity and circadian rhythmicity may be its impact on endoplasmic reticulum (ER) stress. While a healthy ER stress response represents a coping mechanism that allows cellular systems to handle stressful conditions, prolonged or excessive stimuli cause an amplified response that promotes inflammation and hampers insulin signaling [58,59]. Untimely nutrient availability, consequent to a desynchronized meal schedule may, for example, result in an excessive ER stress activation. A dysregulated ER stress response was found to also be associated with circadian misalignment [60]. Interestingly, it has been demonstrated that Sirt1 activation attenuates ER stress in animal and cellular models [59,61], thus hinting at a possible mechanism by which circadian clock control may interact with the regulation of insulin action. 

An equal, or even more convincing, possible candidate is represented by the phosphatase PHLPP. This protein was identified via a rational search of the NCBI database of human genome sequences in 2005 after a year-long quest for a phosphatase specifically targeting the serine/threonine kinase Akt [62]. The first description of this protein dates to a few years earlier, when it was known as SCOP (suprachiasmatic nucleus circadian oscillatory protein) to reflect the observation that its mRNA levels in the suprachiasmatic nucleus, where the master clock resides, oscillate in a circadian rhythm-dependent fashion [63]. SCOP/PHLPP was subsequently demonstrated to be able to specifically dephosphorylate Akt serine 473 residue in vitro as well as in vivo [6,64,65,66]. The protein was then renamed PHLPP, after its domain composition, to highlight its role in the regulation of Akt function [62]. A couple of years later a second member of the family was identified and named PHLPP2 [67]. The two isoenzymes, encoded by two different genes, show high domain similarity but a certain degree of substrate specificity, with PHLPP1 preferentially dephosphorylating Akt2 and Akt3 isoforms and PHLPP2 favoring Akt1 and Akt3 [67]. Subsequent studies showed that, while being highly specific for the serine 473 residue over the threonine 308 motif of Akt, PHLPP phosphatases have several intracellular substrates recurring within the insulin signaling pathway. As summarized in Figure 1, these include conventional and novel protein kinase C (PKC) family members, protein S6 kinase (S6K1) and Mst1/STK4 serine/threonine kinase, implicated in insulin effects on cell survival and proliferation (Figure 1) [6]. 

PHLPP proteins have been found upregulated in skeletal muscle and adipose tissue biopsies from insulin-resistant individuals [64,66] as well as in rat pancreatic beta-cells chronically exposed to glucose toxicity [65]; it is, therefore, possible to hypothesize that they may have a clinically relevant role in the pathogenesis of IR-related diseases. Despite the initial evidence of SCOP/PHLPP1 circadian rhythm-dependent regulation, which is supported by the subsequent observation that PHLPP1-null mice have a drastically impaired ability to stabilize the circadian period after light-induced resetting [68], the hypothesis that PHLPPs may represent the missing link between IR disorders and circadian rhythm alterations has not been experimentally addressed to date. Nonetheless, it is tempting to speculate that circadian oscillations of PHLPP expression may determine daily variations in the activation state of insulin signaling key players, thus affecting cellular metabolic flexibility. 

## 6. Conclusions

As summarized in this review article, there are multiple and complex mechanisms linking IR and desynchronized circadian rhythms. Indeed, not only do central and peripheral circadian clocks have fundamental roles in modulating insulin sensitivity, throughout the plethora of direct and indirect processes that we have attempted to summarize herein, but a healthy insulin response is also required to maintain circadian gene expression rhythmicity [69]. A thorough change of life habits, with the return to synchronized food intake, physical activity, and rest, would doubtless halt this vicious cycle, resulting in a reduced incidence of IR and metabolic diseases. However, such a comprehensive change may be unfeasible and incompatible with the demand of modern society. Thus, clarifying the pathways underlying the impact of circadian rhythmicity disruption on metabolic health may contribute to the identification of therapeutic targets that may be exploited to cure or prevent IR-related diseases.

## Figures and Tables

**Figure 1 ijerph-20-00029-f001:**
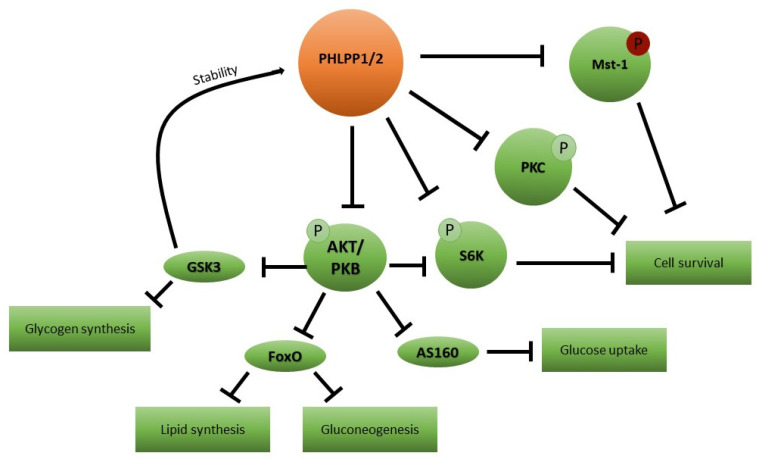
PHLPP1/2 targets in insulin signaling. Schematic representation of the effects of an enhanced PHLPP1/2 expression and or/activity on cell metabolism and survival. PHLPP1/2: PH domain Leucine-rich repeat Protein Phosphatase; Mst-1: macrophage-stimulating 1; P: phosphorylated residues; PKC: protein kinase C; S6K1: ribosomal protein S6 kinase beta-1; Akt/PKB: protein kinase B; AS160: Akt substrate of 160 kDa; FoxO: forkhead box protein O; and GSK3: glycogen synthase kinase-3.

**Table 1 ijerph-20-00029-t001:** Impact of circadian rhythm dysregulation on insulin sensitivity in peripheral tissues. CR: circadian rhythm; GI: gastrointestinal; InsR: insulin receptor; FA: fatty acid; RBP4: retinol-binding protein 4; and JAK/STAT: Janus Kinase and Signal Transducer and Activator of Transcription.

Organ/System	CR Regulated Function	Direct Effect When CR Is Altered	Output
GI tract	Motility	Altered transit time	Impaired insulin signaling due to gluco/lipo toxicity
	Nutrient digestion and absorption	Increased/decreased nutrient flux	Impaired insulin signaling due to gluco/lipo toxicity
	Incretin release	Altered insulin secretion	Hypo/Hyperinsulinemia, InsR downregulation
Liver	Nutrient utilization	Altered FA and glucose flux	Impaired insulin signaling due to gluco/lipo toxicity
Skeletal muscle	Response to physical activity	Impaired energy consumption	Reduced insulin sensitizing potential of physical activity
Adipose tissue	Differentiation	Altered fat depots	Impaired insulin signaling due to inflammation
	Secretion of inflammatory factors	Inflammation	Impaired insulin signaling due to inflammation
	Resistin release	Altered food intake and energy expenditure	Impaired insulin signaling due to gluco/lipo toxicity
	Visfatin release	Increased inflammation	Impaired insulin signaling due to inflammation
	Adiponectin release	Altered energy metabolism	Impaired insulin signaling due to gluco/lipo toxicity
	Resistin release	Increased proinflammatory cytokines secretion	Impaired insulin signaling due to inflammation
Liver/adipose tissue	RBP4 release	Time-dependent activation of JAK/STAT pathway	Impaired insulin signaling due to JAK/STAT induced genes

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
