# Peer review of "Circadian Clock Desynchronization and Insulin Resistance"

_ijerph, 2022, doi:10.3390/ijerph20010029_

Round 1

Reviewer 1 Report

In this review, Catalano and colleagues discuss the current understanding of the molecular and cellular mechanisms underlying the effect of circadian dysregulation on insulin action. Their final conclusion is that a radical change in lifestyle, with a return to synchronised food intake, physical activity and rest, would undoubtedly break the vicious circle linking IR to circadian rhythm dysregulation. However, it is also noted that this is difficult to reconcile with the needs of today's society; clarification of the pathways involved may help to identify therapeutic targets that can be exploited to cure or prevent IR-related diseases.

Comments/suggestions:

1.            In the introduction, it would be worthwhile to briefly state how many people worldwide are affected by insulin resistance and how many are affected by circadian rhythm abnormalities.

2.            For Table 1, I suggest merging the same items in the organ/system column (e.g. merge 3 pieces of GI tract and 6 pieces of adipose tissue separately).

3.            The authors present a number of biochemical pathways in the relationship between circadian rhythm and IR, but their final conclusion is that lifestyle factors are the key. Why did they come to this conclusion?

4.            Are there any data in the available literature that show that the role of certain biochemical/genetic factors is different in certain populations (ethnicities) than in the majority?

5.            What does the "P" in Figure 1 stand for?

The topic is extremely interesting and complex, but it is not possible to judge whether the authors have summarised the results of all the available literature without the criteria typical of a systematic review. I recommend that the authors summarise the topic in the form of a systematic review in the future.

Author Response

Reviewer 1

We thank the Reviewer for the insightful suggestions that have helped us to improve our Review.

We have addressed them as follows:

  1. In the introduction, it would be worthwhile to briefly state how many people worldwide are affected by insulin resistance and how many are affected by circadian rhythm abnormalities.

R1: We thank the Reviewer for this suggestion. We have now included the required information (Introduction lines 31-32 and lines 76-79, new references #3,10)

  1. For Table 1, I suggest merging the same items in the organ/system column (e.g. merge 3 pieces of GI tract and 6 pieces of adipose tissue separately).

R2: We wish to thank the Reviewer for this suggestion, which have helped us to improve table 1 readability

  1. The authors present a number of biochemical pathways in the relationship between circadian rhythm and IR, but their final conclusion is that lifestyle factors are the key. Why did they come to this conclusion?

R3: As reported in the Conclusions section of our manuscript, we believe that the multiple biochemical and intracellular pathways involved in the relationship between alterations of circadian rhythmicity and insulin sensitivity would all take advantage from a resynchronization of daily behaviors (food intake, sleep, physical activity). If the Reviewer has any suggestion on how to render this point more straightforward for potential Readers, we will be more than happy to implement it.

  1. Are there any data in the available literature that show that the role of certain biochemical/genetic factors is different in certain populations (ethnicities) than in the majority?

R4. We thank the Reviewer for this interesting input. There are in fact several studies demonstrating differences in IR incidence in peoples of different ethnicities (see IDF Diabetes Atlas | Tenth Edition) as well as on the impact of variants of genes involved in insulin signaling and/or predisposing to T2D/IR in different populations (see for example Mahajan A, et al. Genome-wide trans-ancestry meta-analysis provides insight into the genetic architecture of type 2 diabetes susceptibility. Nat Genet 46:234–244, 2014; Keaton et al. A comparison of type 2 diabetes risk allele load between African Americans and European Americans. Hum Genet 133(12):1487-95, 2014; Liu et al. Trans-ethnic Meta-analysis and Functional Annotation Illuminates the Genetic Architecture of Fasting Glucose and Insulin. Am J Hum Genet 99:56-75, 2016). There are also a number of studies describing circadian rhythms in subjects with different backgrounds (see for example this recent study from Li et al. Demographic characteristics associated with circadian rest-activity rhythm patterns :a cross-sectional study. International Journal of Behavioral Nutrition and Physical Activity 18:107, 2021) as well as on the prevalence of circadian genes variants in different population (see for example Uemura H et al. Variant of the clock circadian regulator (CLOCK) gene and related haplotypes are associated with the prevalence of type 2 diabetes in the Japanese population J Diabetes 8:667-76, 2016; Garcia-Rios A et al. Beneficial effect of CLOCK gene polymorphism rs1801260 in combination with low-fat diet on insulin metabolism in the patients with metabolic syndrome Chronobiol 31:401-8, 2014; Tangestani H et al. Variants in Circadian Rhythm Gene Cry1 Interacts with Healthy Dietary Pattern for Serum Leptin Levels: a Cross-sectional Study. Clin Nutr Res 10:48-58, 2021). We feel, however, that incorporating these data in our Review may result in a shift of its main focus, which was to illustrate the relationship between CR alterations and IR. We sincerely hope that the Reviewer may share our point-of-view, but are, nonetheless, willing to accept any suggestion that may help us to introduce this important issue in a further revised version of our Review.

  1. What does the "P" in Figure 1 stand for?

R5: We apologize for this oversight. The “Ps” in figure 1 indicate phosphorylated residues; this has now been added in the figure legend.

Reviewer 2 Report

General comments:

The authors have provided a review of circadian clock dysregulation and potential mediators. This is an interesting topic and highly relevant as greater research emerges in this area. Comments have been provided for constructive feedback to help improve the manuscript.

Specific Comments:

Line 15: Spell out PHLPP for first use

Figure: I like the figure, or is it a graphical abstract? However, it needs more detail to convey the point intended. What is happening at liver, GI, etc? Perhaps add arrows to indicate impaired insulin signaling instead of the text box? It just seems to be missing something, I’m not sure how PHLPPs and Sirtuins are meant to be linked in the figure.

Line 31-36: general editing to improve clarity. Can probably remove a few commas, split up second sentence. Particularly in this area but applied to the manuscript in general as well.

Line 44: spell out PKB first use

Line 68-70: should these all be spelled out first use?

Line 86: can you delete ‘per se’?

Table 1. You can delete ‘insulin signaling’ from each box in the last column since it is in the column heading. This would improve the readability of the table. For example, the first box would read Impaired; due to gluco/lipo toxicity. Table is difficult to read as-is.

Check ref 21: * I do not agree with these conclusions, and they do not seem to match the conclusions from the referenced manuscript. “dysreg circadian homeostasis may dampen the insulin sensitizing potential of PA, thus desynch meal and exercise timing would generate self-amplifying vicious cycle that ultimately worsen insulin sensitivity and further impairs metabolic balance.” I don’t see how this creates a vicious cycle where PA would be causing IR. Perhaps PA may lose insulin-sensitizing ability, as stated, but this is not the same as a ‘viscous cycle’. This statement should be reconsidered/rephrased.

Lines 142: I noticed a review was cited here, I suggest either using a primary literature source or indicate this topic has been reviewed by this citation.

Line 292- Was PER1 already mentioned? If not, spell out first use. It would also be helpful to provide context/background for the reader on why this is significant/what it does.

Line 293- 298: not sure this interpretation is correct, seems like clock and bmal are controlling SIRT, therefore would be master regulators. Ref 54?

Line 302-305 – way too many commas, need editing for clarity

Linde 311-351: I think you should provide basic background for the reader on relevance of the pathways mentioned.

Author Response

We wish to thank the Reviewer for the helpful suggestions that have been addressed as follows:

  1. Line 15: Spell out PHLPP for first use

R1 We apologize for this, and the following, oversights. PHLPP has now been spelled at firs use in the abstract (line 15)

  1. Figure: I like the figure, or is it a graphical abstract? However, it needs more detail to convey the point intended. What is happening at liver, GI, etc? Perhaps add arrows to indicate impaired insulin signaling instead of the text box? It just seems to be missing something, I’m not sure how PHLPPs and Sirtuins are meant to be linked in the figure.

R2: We wish to thank the Reviewer for her/his suggestion that have helped us to improve the graphical abstract. We have now modified it and we hope to have made it clearer. We will nonetheless be more than happy to implement any suggestion for further improvements

  1. Line 31-36: general editing to improve clarity. Can probably remove a few commas, split up second sentence. Particularly in this area but applied to the manuscript in general as well.

R3: We have modified the sentences (now at lines 33-38) as well as others throughout the manuscript. We will really appreciate if the Reviewer may indicate additional specific sentences that s/he feels may need to be rephrased for clarity.

  1. Line 44: spell out PKB first use

R4: We have spelt PKB at first use (now at line 46)

  1. Line 68-70: should these all be spelled out first use?

R5: We have spelt all the indicate abbreviations at first use (now lines 70-74)

  1. Line 86: can you delete ‘per se’?

R6: We have deleted “per se” as requested (now line 96)

  1. Table 1. You can delete ‘insulin signaling’ from each box in the last column since it is in the column heading. This would improve the readability of the table. For example, the first box would read Impaired; due to gluco/lipo toxicity. Table is difficult to read as-is.

R7. We wish to thank the Reviewer for the suggestion aimed to improve the readability of Table 1 and we sincerely apologize for not implementing exactly her/his suggestion. However, we feel that the change of the column heading that we have made ameliorates the clarity of the Table and it is more appropriate since not all the effects observed are directly related to insulin signaling. We hope that the Reviewer agrees with our choice but are, nonetheless, ready to perform further corrections that s/he may think necessary. 

  1. Check ref 21: * I do not agree with these conclusions, and they do not seem to match the conclusions from the referenced manuscript. “dysreg circadian homeostasis may dampen the insulin sensitizing potential of PA, thus desynch meal and exercise timing would generate self-amplifying vicious cycle that ultimately worsen insulin sensitivity and further impairs metabolic balance.” I don’t see how this creates a vicious cycle where PA would be causing IR. Perhaps PA may lose insulin-sensitizing ability, as stated, but this is not the same as a ‘viscous cycle’. This statement should be reconsidered/rephrased.

R8 We thank the Reviewer for this suggestion. The sentence has been rephrased (now at lines 137-140) , as suggested.

  1. Lines 142: I noticed a review was cited here, I suggest either using a primary literature source or indicate this topic has been reviewed by this citation.

R9. We are now citing the pertinent primary literature source, as suggested (now ref #24)

  1. Line 292- Was PER1 already mentioned? If not, spell out first use. It would also be helpful to provide context/background for the reader on why this is significant/what it does.

R10: PER 1 has indeed be mentioned before (line 71) and it has been stated that it is involved in transcriptional-translational feedback circuits regulating central and peripheral clocks. If the Reviewer feels that any additional information may be helpful here (now lines 165-167) we will gladly include it.

  1. Line 293- 298: not sure this interpretation is correct, seems like clock and bmal are controlling SIRT, therefore would be master regulators. Ref 54?

R11. We thank the Reviewer for this comment. We actually meant to underlie the fact that SIRT1 is involved in both circadian rhythm and insulin signaling regulation, we have rephrased the sentence to render it clearer (now lines 305-307),

  1. Line 302-305 – way too many commas, need editing for clarity

R12. As stated above R3, we have rephrased several sentences to improve their clarity. We were unfortunately unable to identify the exact sentences the Reviewer meant here, since the lines number seems to have been misplaced.

  1. Linde 311-351: I think you should provide basic background for the reader on relevance of the pathways mentioned.

R13. We have modified the sentences (now at lines 343-345) to improve their clarity. Please note that basic background on insulin signaling has been provided in the introduction and we have also included a figure (Fig.1) to clarify the role of PHLPP substrates. If the Reviewer feels that additional info may be needed, we will be happy to include them.